# A fiber array architecture for atom quantum computing

Xiao Li[1], Jia-Yi Hou[1,2,5], Jia-Chao Wang[1,2], Guang-Wei Wang[1,2], Xiao-Dong He[1,3], Feng Zhou[1,3], Yi-Bo Wang[1], Min Liu[1], Jin Wang [1,3,4], Peng Xu [1,3] ✉ & Ming-Sheng Zhan [1,3,4] ✉

Arrays of single atoms trapped in optical tweezers are increasingly recognized as a promising platform for scalable quantum computing. In both the fault-tolerant and NISQ eras, the ability to individually control qubits is essential for the efficient execution of quantum circuits. Time-division multiplexed control schemes based on atom shuttling or beam scanning have been employed to build programmable neutral atom quantum processors, but achieving high-rate, highly parallel gate operations remains a challenge. Here, we propose a fiber array architecture for atom quantum computing capable of fully independent control of individual atoms. The trapping and addressing lasers for each individual atom are emitted from the same optical waveguide, enabling robust control through common-mode suppression of beam pointing noise. Using a fiber array, we experimentally demonstrate the trapping and independent control of ten single atoms in two-dimensional optical tweezers, achieving individually addressed single-qubit gates with an average fidelity of 0.9966(3). More significantly, we perform simultaneous arbitrary single-qubit gates on four randomly selected qubits, resulting in an average fidelity of 0.9961(4). Our work paves the way for time-efficient execution of quantum algorithms on neutral atom quantum computers.

Owing to the outstanding scalability in qubit numbers[1–6] and reconfigurable Rydberg interactions[7–13], single-atom arrays trapped in optical tweezers serves as a crucial platform for exploring new physics in complex quantum systems[14–18], while also being a prominent candidate for quantum computing[19–36]. Recent experiments have demonstrated logical qubit encoding and logical gate operations on hundreds of moving atoms[21], emphasizing the notable advantages of the atom array platform in terms of qubit count, gate operation parallelism, and qubit connectivity. However, achieving practical quantum computing still requires significant progress in reducing error rates and increasing clock speeds[37–39]. Thus, one of the upcoming key challenges for atom quantum computing is to develop an architecture that supports fast, highly parallel, scalable and stable addressing operations.

A natural way to achieve programmable quantum computing on this platform is to first use optical diffraction devices, such as spatial light modulators (SLMs)[3] or acousto-optic deflectors (AODs)[2], to create large qubit arrays, and then incorporate individual addressing capabilities into these arrays. In recent years, two schemes based on time-division multiplexed optical addressing have been demonstrated to realize this architecture. One involves multiplexing the addressing light by shuttling atoms, where qubit addressing is achieved by shifting qubits in and out of a large, uniform driving laser[19,21]. This facilitates

[1]Division of Precision Measurement Physics, Wuhan Institute of Physics and Mathematics, Innovation Academy for Precision Measurement Science and Technology, Chinese Academy of Sciences, Wuhan, China. [2]School of Physical Sciences, University of Chinese Academy of Sciences, Beijing, China. [3]Wuhan Institute of Quantum Technology, Wuhan, China. [4]Hefei National Laboratory, Hefei, China. [5]Present address: CAS Cold Atom Technology (Wuhan) Co. Ltd, Wuhan, China. ✉e-mail: xupeng@apm.ac.cn; mszhan@apm.ac.cn

parallel qubit operations and enables on-demand, non-local qubit connections. However, moving qubits spatially, especially between different trap sites, will significantly increase the idle time between qubit gate operations, typically reaching several hundred microseconds. Another approach uses an AOD to steer tightly focused addressing beams onto a static single-atom array, reducing the idle time to sub-microseconds[20,22]. However, this scheme favors sequential operations and faces challenges in maintaining precise and stable alignment of the addressing light with the qubits. Additionally, both approaches use AODs to achieve fast atom movement or quick beam steering, which restricts the simultaneously addressed qubits to a square grid pattern. In the long term, executing deep quantum circuits on a large number of qubits is necessary for practical error-correcting quantum algorithms. This requires high gate rates and highly parallel operations, considering finite coherence times and constraints on acceptable execution times[37]. In the NISQ era, high flexibility in qubit implementation is crucial for achieving quantum advantage, such as in random quantum circuit sampling[40,41]. Several other addressing schemes have been proposed, but fully meeting the numerous requirements remains challenging[42–44].

To overcome these challenges, we propose a fiber array architecture to independently control single-atom qubits in atom arrays for quantum computing. Each fiber channel is connected to an optical module that integrates both trapping and addressing laser beams, which are focused on the same location through identical optical paths. The number of qubits and control signals can be scaled directly by replicating the optical modules and fiber channels. This design offers the advantage of inherent spatial alignment between each addressing beam and its corresponding qubit, while enabling precise and full control over individual atoms. Experimentally, we used a fiber array to trap and independently control 10 single atoms, demonstrating individually addressed single-qubit gates with an average fidelity of 0.9966(3). Notably, we also achieved parallel implementation of arbitrary single-qubit gates on this 2D atom array, a task previously considered highly challenging. Moreover, we successfully demonstrate Rydberg blockade between two individual atoms in fiber array traps. Our work presents a robust method for the independently control of qubits in a two-dimensional single-atom array. Combined with high fidelity addressed long-range Rydberg gates[22,38,39], this offers a feasible architecture for executing quantum algorithms on static single-atom qubits with high time efficiency.

## Results
### Scheme
In our previous design[45], lasers emitted from a one dimensional waveguide array were directly focused by a microlens array, thereby creating an array of light spots. In order to allow adjacent single atoms to enter the Rydberg blockade region, the spacing between the optical waveguides needs to be reduced to just a few micrometers, leading to substantial crosstalk between waveguides. Moreover, etching microlenses at the ends of optical waveguides introduces considerable aberrations.

In this work, we utilize a 2D fiber array to provide transmission channels for both trapping light and addressing light. The basic architecture is shown in Fig. 1a. The light field, emitted from a fiber array with spacing of several tens of micrometers, are projected inside a vacuum chamber with a reduced image, forming an array of optical traps spaced just a few micrometers apart. This configuration relaxes the spacing requirements between optical waveguide channels to tens of micrometers. Moreover, optical aberrations during beam focusing

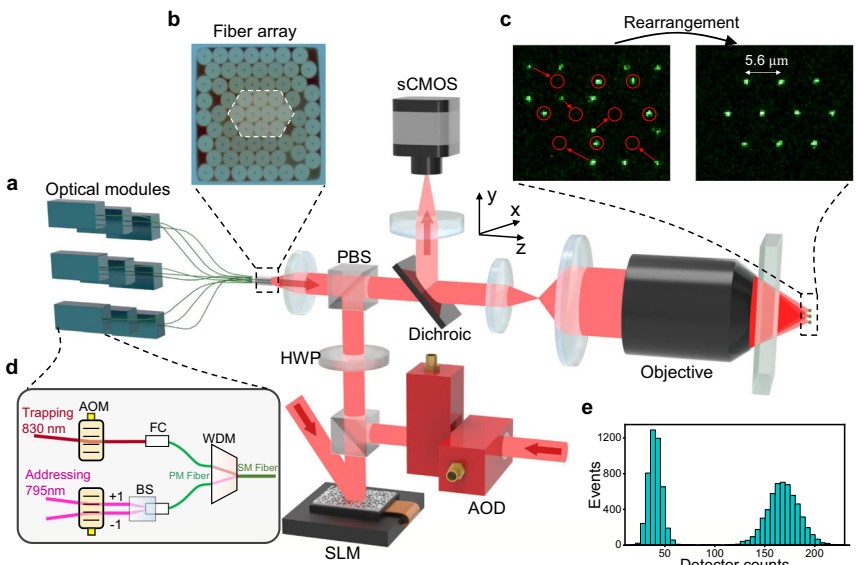

**Fig. 1 | Experimental scheme. a** Basic experimental setup for the trapping, rearrangement, manipulation, and detection of single-atom arrays. The system includes a steerable optical tweezer generated by an acousto-optic deflector (AOD) and two static optical tweezer arrays: one created by an optical fiber array, and the other by a spatial light modulator (SLM). The SLM-generated traps serve as a reservoir for atom rearrangement, whereas the fiber-array-generated traps act as the main register for qubit operations. Each optical module controls an individual qubit confined in a fiber trap by delivering tightly focused trapping (830 nm) and addressing (795 nm) beams through a shared optical path. The beams are emitted from the fiber array end face, and focused into the vacuum chamber by a high-NA (NA = 0.7) objective. Atom fluorescence is collected through the same objective and detected by an sCMOS camera after reflection from a dichroic mirror. **b** The cross-section of the fiber array. The 10 fibers enclosed in the dashed box are chosen for this experiment. **c** Single-shot fluorescence image of an atom array with 50 ms exposure time. Left: randomly loaded array (red circles indicate positions of fiber traps). Right: defect-free array obtained after atom rearrangement, with the SLM traps turned off prior to the second exposure. **d** A schematic showing optical setup in the optical module. A single 830 nm trapping beam and a pair of 795 nm addressing Raman beams each pass through an AOM before being combined into the same SM fiber using a WDM (see Supplementary Information). **e** Histogram of collected photons for one of the fiber traps during the initial loading process. Two distinct peaks indicate the presence of one atom (right peak) and no atom (left peak) in the trap. PBS, polarizing beam splitter; HWP, half-wave plate; AOM, acousto-optic modulator; FC, fiber coupler; BS, beam splitter; WDM, wavelength-division multiplexer.

can be effectively corrected with the advanced optical design techniques. Similar methods based on 1D waveguide arrays have recently been applied to single-ion addressing in linear ion traps. However, the ion trap and the addressing light are separate, each subject to its own positional fluctuations.[46–49].

The experimental setup is shown in Fig. 1a. We customize a 2D fiber array consisting of 64 single-mode fibers (Nufern 780-HP fiber), with the cladding at the end of each single-mode fiber reduced to 40 μm. In addition to atom trapping, this fiber array also enables individual addressing within the atom arrays. The trapping and addressing light for each single atom are contained within a single optical module, as shown in Fig. 1d.

The 10 trapping laser beams are individually coupled into 10 selected single-mode fibers arranged in a hexagonal pattern and emit from the end face of the fiber array, Fig. 1b, forming 10 Gaussian beams with a spacing of 40 μm. The magnification of the optical system is 0.14, so the spacing of the optical trap array is around 5.6 μm. Individual $^{87}$Rb atoms are randomly loaded from a 3D-MOT into two static optical tweezer arrays, one generated by an SLM and the other by a fiber array, with an average loading probability of 0.55. These atoms are subsequently rearranged using a steerable tweezer into a defect-free array aligned with the 10 fiber traps, achieving a single-shot success probability of 0.76, Fig. 1c.

## Individual addressing of single-atom qubit arrays

We first demonstrate individual addressing of this atom array. Figure 2a shows the relevant atomic-level diagram. In order to independently control 10 qubits, we directed 10 separated addressing Raman laser beams into the 10 single-mode fibers. Individual and parallel manipulation of arbitrary single-atom qubits can be achieved by controlling the amplitude, frequency, and phase of each addressing laser.

In our architecture, the trapping and addressing light share the same optical path after emerging from the fiber array. While this configuration enhances the stability and flexibility of addressing control, it introduces challenges in differentiating the beam parameters for the trapping and addressing light.

A primary concern is the polarization of the laser beams, which must be adapted to meet the varying requirements of different experimental stages, including atom loading, polarization gradient cooling (PGC), site-selective control, and imaging. We address this issue by dynamically adjusting the laser polarization using a liquid crystal variable retarder (LCVR). This setting also enables magic-intensity trapping, thereby enhancing the coherence time of the single-atom qubits[24].

Another challenge is the non-uniform intensity experienced by atoms due to the tightly focused addressing beam[50]. To mitigate this issue, we place a 1 mm central aperture filter before the dichroic mirror to reduce the numerical aperture of the addressing beam while leaving the trapping light largely unaffected (see Supplementary Information).

Individual addressing on any single atom can be achieved by switching on the corresponding Raman laser beam. The upper part of Fig. 2b exemplarily shows the Rabi oscillation between $|0\rangle$ and $|1\rangle$ on one single-atom qubit, with a fitted Rabi frequency of $\Omega_{addressed} = 2\pi \times 34.6$ kHz. The lower part displays the Rabi oscillations of qubits in other traps over an extended evolution time, where no distinct oscillatory behavior is observed. We extract the Rabi rate crosstalk (defined by $\Omega_{adjacent}/\Omega_{addressed}$[46,51]) on all other unaddressed single-atom qubits is less than 0.1%. Similarly, crosstalk for other target single atoms is measured, resulting in a maximum Rabi rate crosstalk of 1.0% (see Supplementary Information).

We evaluate the addressed single-qubit gates across the entire single-atom array, site by site, using the well-established Randomized Benchmarking (RB) method[27–30]. Single-qubit RB data for one site are displayed in the upper panel of Fig. 2c. The fidelity of addressed single-

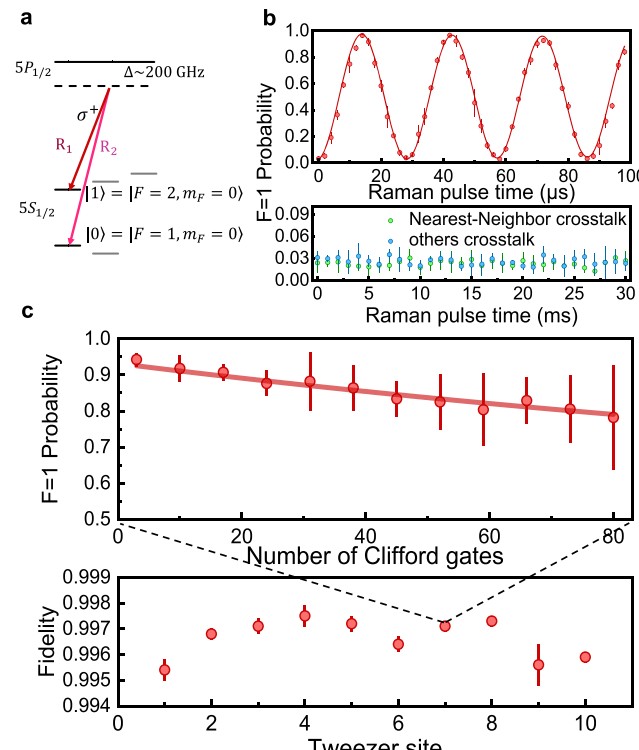

**Fig. 2 | Individual addressing of single-atom qubit arrays. a** Energy level schematic for the qubits. Qubits are encoded in the two hyperfine ground states of $^{87}$Rb atoms: $|0\rangle \equiv |5S_{1/2}, F = 1, m_F = 0\rangle$ and $|1\rangle \equiv |5S_{1/2}, F = 2, m_F = 0\rangle$, which are coupled by a two-photon Raman transition with a single-photon detuning of $2\pi \times 200$ GHz relative to the $5P_{1/2}$ manifold. **b** Rabi oscillation of the 10 single atoms when addressing one single atom. The upper panel displays the Rabi oscillations of a specific qubit (qubit 7), driven by a Raman pulse, whereas the lower panel depicts the evolution of the other qubits subjected to a longer Raman pulse. **c** Randomized benchmarking (RB) of individually addressed single-qubit gates. The upper panel shows results for qubit 7, with a fidelity of 0.9971(2). The lower panel displays a site-by-site analysis of average single-qubit fidelities, varying between 0.995 and 0.998. All error bars represent one standard deviation.

qubit gates at each site is summarized in the lower panel of Fig. 2c, with the fidelity across the entire array ranging from 0.995–0.998 and an average fidelity of 0.9966(3). While a detailed error analysis is not performed, for our current experimental setup, gate errors mainly come from two sources: (1) Spatial variation in the Rabi frequency experienced by single atoms, and (2) the disruption of qubit coherence due to the spatially non-uniform differential AC Stark shift on hyperfine states of trapped single atoms, induced by addressing light. They can both be effectively suppressed by lowering the temperature of the single atoms in the future.

## Parallel addressing of arbitrary single-atom qubits

To demonstrate the ability in parallel manipulation of qubits, we first perform a simultaneous Ramsey experiment on the entire atom array. We set unique detunings and phase offsets for the Ramsey sequence at each site, as shown in Fig. 3a, enabling the qubits to exhibit distinct oscillation frequencies and initial phases, as clearly demonstrated in the results shown in Fig. 3b.

Next, we perform parallel addressing of arbitrary single-atom qubits. To begin, we present parallel Rabi oscillation in arbitrary single atoms. Four addressing beams are simultaneously applied to four randomly selected single atoms, which are arranged in an irregular pattern as shown in Fig. 4a. By scanning the duration of the addressing laser pulses, we obtain the Rabi oscillations of the four target qubits.

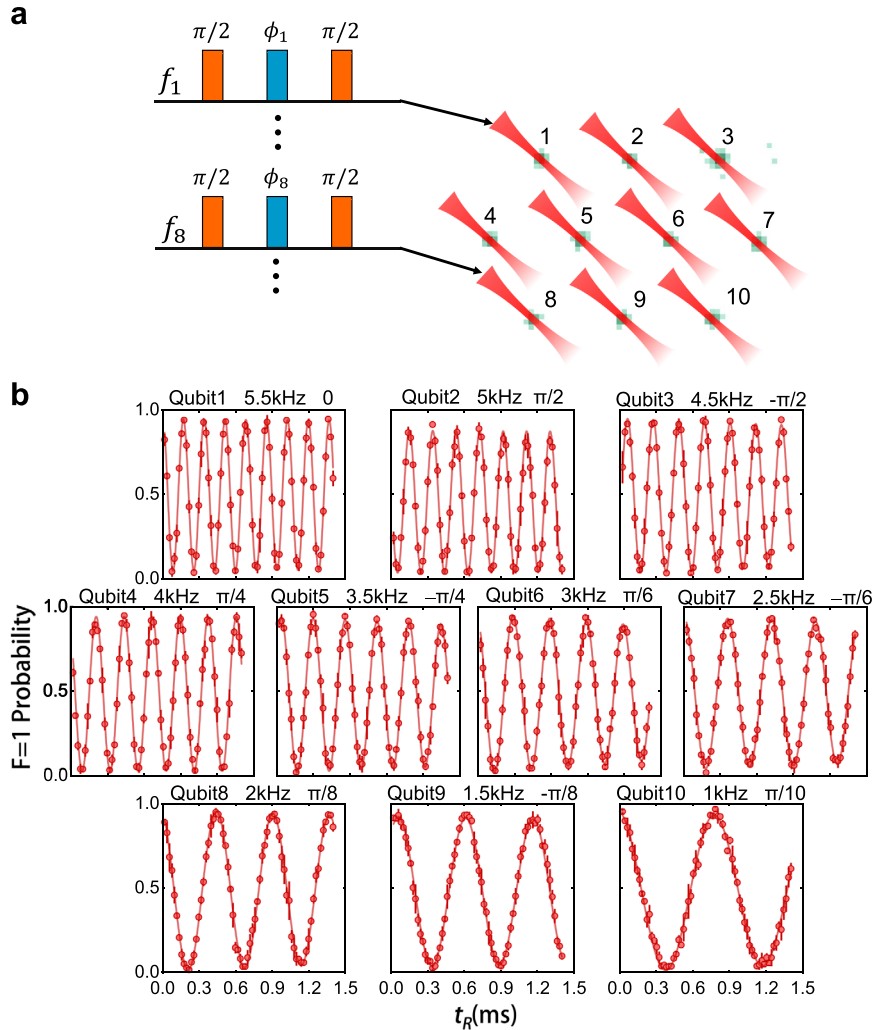

**Fig. 3 | Simultaneous Ramsey experiment on the entire array. a** The Ramsey sequence comprises two π/2 pulses with a phase-shift pulse inserted between them to alter the phase of the second π/2 pulse. At each site, the Ramsey sequence is configured with unique two-photon frequencies $f_i$ and phase offsets $\phi_i$. **b** 10 single-atom qubits undergo simultaneous Ramsey oscillations, each manifesting distinct frequencies and initial phases. All error bars represent one standard deviation.

The dephasing rates of the Rabi oscillations are significantly faster than anticipated, primarily due to spatial interference among the addressing lights. In our experiment, since all the addressing lights originate from a single laser, they are coherent with one another. Crosstalk among addressing beams, resulting from optical aberrations or surface scattering, will produce time-varying effects on the light intensity at the addressed site, as also noted in refs. 20,22. The rate of intensity fluctuations depends on the rate of relative phase drift among different addressing beams. In our current setup, it is much slower than the Rabi rate but is comparable to the event-counting rate (-1 Hz in our experiment). From the Rabi oscillation in Fig. 4a, we numerically extract that the slow modulation amplitude of the Rabi frequency ranges from 2% to 5%. This leads to per-gate error for random Clifford gates between $3.6 \times 10^{-4}$ and $1.2 \times 10^{-3}$, which have a minimal effect on our current gate fidelity.

Then, we perform simultaneous RB experiments on four targeted qubits by applying an independent sequence of Clifford gates to each qubit. Results are displayed in Fig. 4b, from which the fidelity of the parallel-addressed single-qubit gates for each qubit is extracted. The resulting fidelities range from 0.995–0.997, closely approximating the fidelity of individual-addressed single-qubit gates, which confirms that the variation in the Rabi frequency is a slow modulation.

## Rydberg blockade between two individual atoms

Finally, we perform Rydberg excitation experiments on this atom array and clearly observe Rydberg blockade between two neighboring atoms. The excitation from $|1\rangle$ to $|r\rangle = |68D_{5/2}, m_j = 5/2\rangle$ is realized via a counterpropagating two-photon transition using 780 nm ($\sigma^+$) and 480 nm ($\sigma^+$) lasers, both frequency-stabilized to a high-finesse reference cavity[32]. As depicted in Fig. 5a, the 780 nm beams are coupled into the fiber channels of atoms 5 and 6 via fiber WDMs, while the 480 nm beam is shaped by an SLM to produce two focused spots with ~4.5 $\mu$m waist aligned to the respective atoms. Figure 5b and c present the Rabi oscillations for individual addressing of atom 5 and atom 6, respectively. The results indicate negligible crosstalk on the adjacent site, and the extracted Rabi frequencies are 2π × 1.16(1) MHz and 2π × 1.19(1) MHz.

When both 780 nm beams are simultaneously applied, we observe a near-perfect Rydberg blockade (Fig. 5d), despite the relatively modest interaction strength of 2π × 24.4 MHz. In this case, the total single-atom excitation probability exhibits coherent Rabi oscillations at $\Omega' = 2\pi \times 1.64(2)$ MHz, with a contrast of about 0.6. This frequency closely matches the expected $\sqrt{2}\Omega$ enhancement, indicating that the two-atom system undergoes coherent evolution between the state $|11\rangle$ and the entangled state $\frac{1}{\sqrt{2}}(|1r\rangle + |r1\rangle)$[52]. Although residual technical imperfections (such as stray electric fields, laser intensity fluctuations, and insufficient atomic cooling) currently degrade the excitation

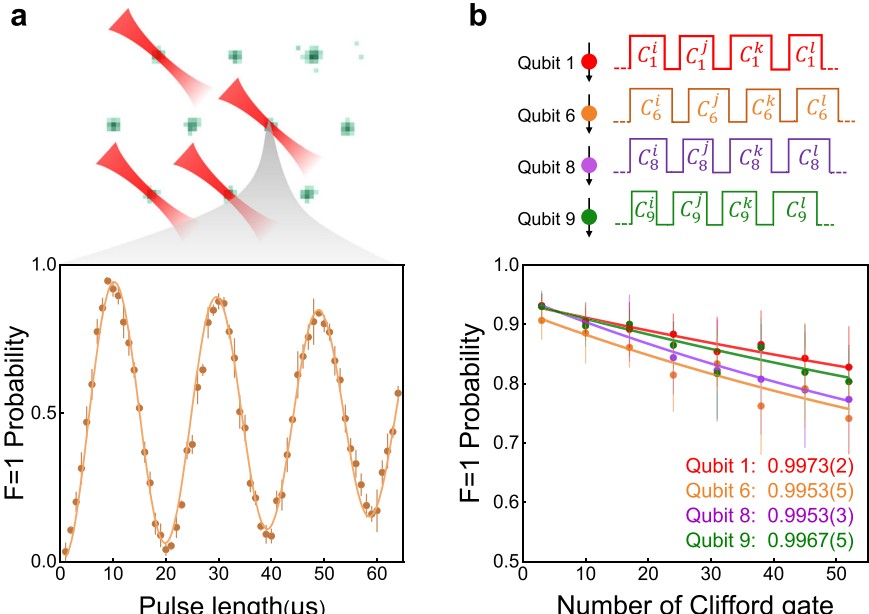

**Fig. 4 | Parallel RB on four arbitrary qubits. a** Parallel addressing of four arbitrary single-atom qubits. Bottom: the Rabi oscillation of qubit 6, which exhibits rapid dephasing. **b** Simultaneous RB experiments on four targeted qubits by applying an independent sequence of Clifford gates to each qubit. The fidelity of parallel-addressed single-qubit gates ranges from 0.995–0.997, closely approximating that of individually addressed single-qubit gates. All error bars represent one standard deviation.

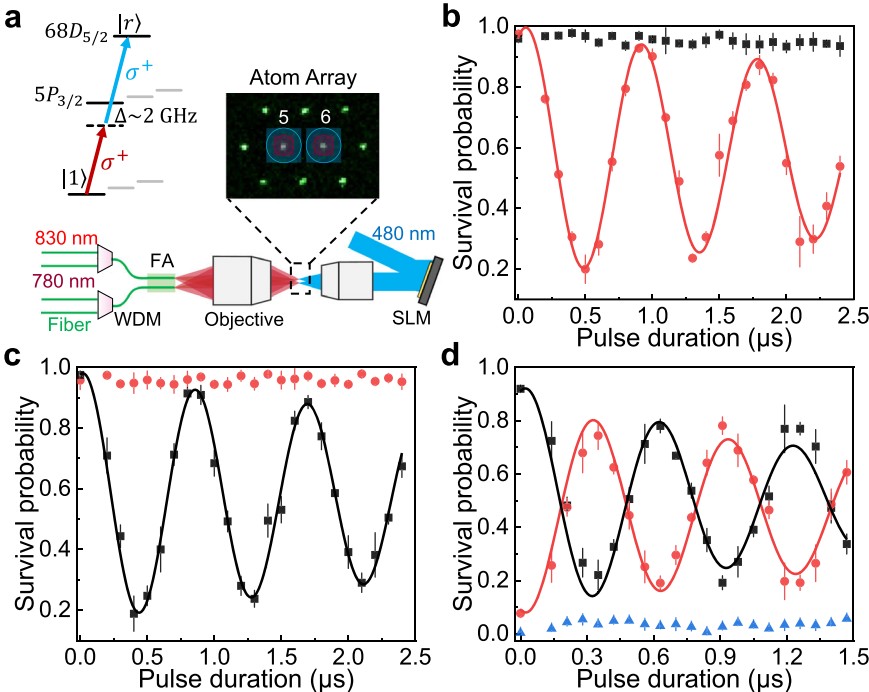

**Fig. 5 | Rydberg excitation and blockade between two individual atoms. a** Atomic-level diagram of the Rydberg excitation scheme and optical layout for delivering 780 nm and 480 nm beams to atoms 5 and 6. WDM, wavelength-division multiplexer; FA, fiber array; SLM, spatial light modulator. **b**, **c** Rabi oscillations under individually addressed Rydberg excitation of atom 5 (**b**) and atom 6 (**c**), with negligible crosstalk observed on the neighboring atom. Red circles and black squares indicate the survival probabilities of atom 5 and atom 6 after Rydberg excitation, respectively. **d** Collective Rabi oscillations under simultaneous Rydberg excitation of both atoms, indicating coherent evolution between $|11\rangle$ and the entangled state $\frac{1}{\sqrt{2}}(|1r\rangle + |r1\rangle)$, as well as a clear signature of Rydberg blockade. Black squares denote events where both atoms remain trapped, red circles correspond to single-atom survival events, and blue triangles indicate simultaneous loss of both atoms. All error bars represent one standard deviation.

efficiency and ground–Rydberg coherence, the present results demonstrate that the fiber array platform provides the essential physical ingredients for realizing high-fidelity two-qubit Rydberg gates[22,31–33].

## Discussion

In conclusion, we demonstrate trapping an array of 10 single atoms in a fiber-array-generated optical tweezers. Based on this platform, we further demonstrate high-fidelity individually and parallelly addressed single-qubit gate operations, as well as a clear Rydberg blockade between two adjacent atoms. Our ability to simultaneously implement different single-qubit gates on arbitrary qubits is extremely helpful for improving the efficiency of quantum circuit execution.

Future works will focus on improving the fidelity of individual-addressed single-qubit gates by further lowering the temperature of single atoms (using advanced cooling techniques such as Raman-sideband-cooling[53,54] or gray-molasses-cooling[55]) and employing addressing light that has a reduced dephasing impact on hyperfine qubits. Further improvements in the fidelity of parallel-addressed single-qubit gates will, however, be limited by crosstalk. Three straight-forward approaches can significantly minimize crosstalk errors: firstly, by reducing the addressing beam's waist radius to lower the intensity at the edges of the Gaussian beam near adjacent sites; secondly, by carefully correcting optical system aberrations and reducing photon scattering on optical surfaces; and thirdly, either by offsetting the carrier frequencies of addressing beams in different channels well beyond the Rabi frequency, or by locking their relative phases, to suppress slow drifts in the Rabi rate caused by optical interference[22]. In parallel, we will also aim to optimize the experimental conditions to enable high-fidelity, individually addressing two-qubit gate on this platform.

Whereas SLMs[3] and AODs[2] are commonly used to generate optical tweezer arrays for trapping atoms, our approach maximizes the inherent advantages of optical tweezers for qubit addressing by ensuring that tightly focused addressing beams are naturally co-aligned with the qubits in space. For optically addressed qubits, excluding shuttling schemes, individual addressing requires a tightly focused laser beam precisely aligned with the target qubit. And single-atom qubits are obtained through the dipole potential of tightly focused trapping lasers, a feature that is distinct from other optically addressed qubits such as ions and NV centers[56,57]. Our method combines the free-space optical paths of the trapping and addressing beams, taking the advantage that each needs to be tightly focused. In this way, the addressing light is inherently aligned with the qubit in space. The accuracy of alignment is only determined by the aberrations of the optical system and is not affected by drifts of the opto-mechanical components. Additionally, although not demonstrated in this work, the 2D optical tweezers generated by our scheme can be quickly and individually turned off, which is highly beneficial for performing deep quantum circuits on a Rydberg atom array platform[20,21].

Admittedly, this proof-of-principle demonstration is carried out on a small scale; the number of qubits can be increased by simply replicating the optical modules at the input end of the fiber array. In our current setup, these modules are built using bulk components. However, to reach the scale required for fault-tolerant quantum computing, it is essential to adopt well-established and scalable integrated photonics solutions[58,59]. For instance, thin-film lithium niobate (TFLN) chips already support functions like beam splitting, amplitude modulation, and phase modulation on a single chip[60–62]. They offer performance close to that of bulk components, but with a smaller size and lower power consumption[63,64]. The number of atoms simultaneously addressable in a single processor is ultimately limited by the number of fiber channels integrated into a single fiber array. To the best of our knowledge, current state-of-the-art fabrication techniques can support thousands of channels with precise and regular spacing[65]. Alternatively, 3D waveguide arrays fabricated via femtosecond laser writing offer a promising path forward[66–68]. To scale toward 100,000 qubits or beyond, quantum networking can be employed to link multiple processors via heralded entanglement[69,70]. Notably, the multi-channel photon-atom interface inherent in the fiber array architecture can significantly enhance the entanglement rate[71,72].

## Methods

### Experiment details

In our setup, to improve the stability of the optical setup and reduce its volume, we employ adhesive bonding techniques to construct 10 optical modules. Additionally, the fiber array is mounted on a fixed rail to ensure the polarization stability of the laser transmitted within it.

Single-qubit gate experiments begins with single $^{87}$Rb atoms being randomly loaded into a two-dimensional optical tweezer array. The loading depth for each trap is 1.4 mK. The temperature of the single atoms is reduced to 18 $\mu$K after PGC. Next, with a $B \approx 3$ G magnetic field (co-aligned with the direction of the dipole trap), the single atoms are pumped into $|1\rangle$ by π-polarized optical pumping light. Following this, we lower the trap depth to 200 $\mu$K and switch the polarizations of the dipole and addressing lights to $\sigma^+$ via an LCVR. The frequencies of the trapping lasers at different sites are offset by more than 4 MHz to suppress atom heating and state leakage caused by interference between the trapping beams[73].

For the Rydberg excitation experiments, we deterministically fill sites 5 and 6 with single atoms by employing atom rearrangement prior to the experimental sequence. A magnetic field of -10 G is applied to resolve the Zeeman sublevels of the $|68D_{5/2}\rangle$ Rydberg state. The Rydberg excitation is carried out during a 4 $\mu$s trap-off window. After the excitation pulse, the traps are turned back on to detect Rydberg atoms. Due to the repulsive (anti-trapping) potential experienced by Rydberg states, Rydberg population is converted into atom loss, which is then measured by subsequent fluorescence imaging.

### Atom rearrangement

To achieve deterministic preparation of defect-free atom arrays in the 10 fiber traps, a spatial light modulator (SLM) and a pair of crossed acousto-optic deflectors (AODs) are integrated into the setup. The SLM generates a 5 × 4 optical tweezer array to capture reservoir atoms, while the AOD produces a steerable optical tweezer for transferring these reservoir atoms into initially unoccupied fiber traps. The SLM and AOD beams are first combined via a polarizing beam splitter (PBS), passed through a half-wave plate (used to adjust the relative power), and then merged with the fiber array beams through a second PBS. Firstly, an image is acquired to identify the initial random distribution of atoms. Subsequently, a rearrangement sequence is computed based on a straightforward strategy that fills each empty trap with the nearest available atom, and delivered to the AOD via an arbitrary waveform generator (AWG), enabling the transfer of atoms from the reservoir into the initially empty fiber traps.

### Randomized benchmarking of single-qubit gates

In the RB experiments, we generate 10 different random sequences of Clifford gates, truncated at various lengths. Each Clifford gate is generated from fundamental gates I, $R_{x,y,z}(\pm \pi/2)$, $R_{x,y,z}(\pi)$. Here, the Z gate is virtually implemented by switching the phase of the Raman laser[30]. In the RB experiments, we generate 10 different random sequences of Clifford gates, truncated at various lengths. After each sequence, a correction gate is applied to rotate the qubit to $|0\rangle$. The fidelity of the addressed single-qubit gate can be extracted by fitting the equation below:

$$\bar{F} = \frac{1}{2} + \frac{1}{2}(1 - d_{if})(1 - 2\varepsilon_g)^\ell, \qquad (1)$$

where $\bar{F}$ is the average probability of $|0\rangle$ state, $\varepsilon_g$ represents the average error per Clifford gate, $d_{if}$ represents the depolarization probability associated with state preparation, $\ell$ denotes the number of random gates applied to the target qubit.

## Data availability

The data that support the findings of this study are available in the Figshare repository at https://doi.org/10.6084/m9.figshare.28193546 (see Ref. 74). Source data are provided with this paper.

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

## Acknowledgements

This work was supported by the National Key Research and Development Program of China under Grant No. 2021YFA1402001 (P.X.), the National Innovation Program for Quantum Science and Technology of China under Grant No. 2023ZD0300401 (M.L.), the National Natural Science Foundation of China under Grants No. 12004397 (X.L.), No. 12261131507 (P.X.), No. 12074391 (P.X.), No. U22A20257 (X.-D. H.), No. 12121004 (X.-D. H.) and No. 12241410 (M.-S. Z.), the CAS Project for Young Scientists in Basic Research under Grant No. YSBR-055 (X.-D.H.),the Natural Science Foundation of Wuhan under Grant No. 2024040701010063 (P.X.), the Major Program (JD) of Hubei Province under Grant No. 2023BAA020 (P.X.).

## Author contributions

X.L., P.X. and M.-S.Z. designed the experiment. X.L., J.-Y.H., J.-C.W., G.-W.W. and P.X. carried out the experiment. X.L., J.-Y.H., J.-C.W., X.-D.H., F.Z., Y.-B.W., M.L., J.W. and P.X. analyzed the data. X.L., J.-Y.H., J.-C.W., P.X. and M.-S.Z. wrote the manuscript. All authors discussed the results and approved the final version of the manuscript.

## Competing interests

The authors declare no competing interests.
