## [Transparent Peer Review file · Nature Communications]

A fiber array architecture for atom quantum computing

Corresponding Author: Professor Peng Xu

Version 0:

Reviewer comments:

Reviewer #1

(Remarks to the Author)

The authors present a system to trap and perform single qubit gates on Rubidium atoms using a fibre array. Optical modules for individual optical channels generate switchable trap light at 830 nm and Raman light at 795 nm. Trapping and Raman light is then from each module coupled into a single fibre within a fibre array. The array is imaged into a vacuum chamber, where Rb atoms are loaded from a MOT using the 830 nm trapping light. The authors then show that they can perform single qubit rotations with the Raman light that propagates together with the trapping light inside of the fibre.

Please find my questions and comments below:

The problem that is addressed is of current interest. Scalable local addressing of a large number of atoms is a central bottleneck in cold atom quantum computation.

I appreciate the method the authors employ to reduce the NA of the addressing light with a dichroic that has an aperture. However, this also means a substantial power loss for the Raman beam.

The authors' current system does not seem to feature a way to resort a the 55% filled array into a dense array of atoms. While this is not crucial for the current demonstration, it would be necessary to scale up.

A problem of this approach, which the authors recognise lies in the scalability arguably cold-atom based quantum computation will need to approach 10,000-100,000 atoms to achieve fault tolerant quantum computation. It does not seem feasible to build optical modules for each channel with bulk components at that scale.

A further problem may arise when we consider two-qubit gates: For high fidelity blockade gates [1] the separation between atoms needs to be on the order of a few μm . It is not entirely clear that very high fidelity blockade gates together with single qubit gates can easily be realised in a static, equidistant array, since for single qubit gates the beam diameter in the atom plane has to be large compared with the trap size to reduce intensity variation induced error. For the narrow atom spacing required for high fidelity blockade gates this would lead to cross-talk.

For these concerns I am not sure that this work reaches the threshold for publication in Nature Communications.

[1]Evered, Simon J., et al. "High-fidelity parallel entangling gates on a neutral-atom quantum computer." Nature 622.7982 (2023): 268-272.

Reviewer #2

(Remarks to the Author)

In this article, the author introduced an innovative approach to implementing the single qubit gate using a fiber array architecture. The study addresses key challenges in achieving efficient and parallelized quantum operations on individual quantum bits (qubits).

In their experiment, the author has experimentally implemented successfully to load 10 Rb atoms into a fiber-generated optical tweezer, and achieve an average gate fidelity of 99.66%. A key advantage is that the parallel operation of the atom qubit in the setting. They have demonstrated, with an example, the simultaneous single-qubit operations on four randomly

selected qubits, with an average gate fidelity of 99.61%. This result highlights the potential for scaling up quantum operations to larger systems. The success of this research opens new avenues for advancing quantum computing technologies, particularly in achieving high-fidelity gate operations on individual qubits and parallel gate operations. Therefore I think the work can be published in NC.

I have a few comments.

1. In the experiment, the author has focused on the single qubit operation. It is nice to see the high fidelity single qubit operation. I wonder why there is no demonstration of the entangled gates (such as two qubit control gates). Will the interference of the laser affect the performance of two qubit gates?

2. Towards the end of the "Scheme" section, the author claimed that the experiment achieved a loading rate 0.55. What is the unit here?

3. Although the results are very interesting, I feel that the Results section contains too many technical details. Some of them might be useful to understand the experiment. However the main aim is still to achieve single and potential multiple qubit gate operation. I would suggest to move some of the subtle details to the SM, such that readers can understand the important advantage of the article, as well as what can be done with this setting.

Version 1:

Reviewer comments:

Reviewer #1

(Remarks to the Author)

The authors have mostly addressed my concerns and have expanded the data taken significantly. I am happy to recommend publication.

I would further recommend to also add the following citation on lithium niobate integrated photonics:
<https://www.nature.com/articles/s41467-024-55423-3>.

Reviewer #2

(Remarks to the Author)

The authors have made substantial improvements in the revised manuscript. All my concerns have been successfully addressed. The authors have clarified the text, and rewritten the manuscript to improve the readability. As far as I can tell, the authors have also addressed the comments of the first referee. Additional discussion and materials have been added to the revised manuscript, which do not only clarify the experimental results, but also highlight the new achievements. Among the changes, I am particularly surprised by the new section on the demonstration of the Rydberg blockade (in section Rydberg blockade between two individual atoms). It has been shown that Rydberg blockade can be realized with the fiber array architecture with a high fidelity. This perfectly reflects the key theme of the article, i.e. a new possible way to carry out quantum computing with the Rydberg atoms.

This is a nice piece of work providing a new way to build neutral atom quantum computers. I believe this work has important and direct contributions to the quantum computing community. It also impacts widely to the study of quantum simulation, light-matter interaction, and quantum information sciences.

I strongly recommend the publication of this work in Nature Communications.

Reply to Referees

We sincerely thank all referees for their valuable time and effort in reviewing our manuscript entitled *A fiber array architecture for atom quantum computing*. We have seriously taken all comments and suggestions, and have made a thorough revision accordingly. Below, we provide a point-by-point response to all questions and suggestions raised by the referees. These comments have helped us further enhance the overall clarity and quality of the manuscript. All changes are clearly highlighted in the **marked version** of the revised manuscript provided.

With best regards,
Peng Xu

Reviewer #1 (Remarks to the Author):

The authors present a system to trap and perform single qubit gates on Rubidium atoms using a fibre array.

Optical modules for individual optical channels generate switchable trap light at 830 nm and Raman light at 795 nm.

Trapping and Raman light is then from each module coupled into a single fibre within a fibre array. The array is imaged into a vacuum chambre, where Rb atoms are loaded from a MOT using the 830 nm trapping light. The authors then show that they can perform single qubit rotations with the Raman light that propagates together with the trapping light inside of the fibre.

Please find my questions and comments below:

The problem that is addressed is of current interest. Scalable local addressing of a large number of atoms is a central bottleneck in cold atom quantum computation.

Response:

We sincerely thank the reviewer for the careful reading and thoughtful evaluation of our manuscript, as well as for the positive assessment of the research topic. The concerns raised by the reviewer are well aligned with our future direction. Below, we have responded to the reviewer's comments point by point, and we hope that our replies and the corresponding revisions will satisfactorily resolve the raised concerns.

Q1. I appreciate the method the authors employ to reduce the NA of the addressing light with a dichroic that has an aperture. However, this also means a substantial power loss for the Raman beam.

Response:

We thank the reviewer for recognizing our approach for reducing the NA of the addressing beam, and we agree with the concern regarding Raman beam power loss.

Experimental measurements in our current setup indicate that the 1 mm aperture leads to approximately 80% power loss of the Raman beam.

Nevertheless, Raman power utilization can be significantly improved by relaxing the aperture constraint via optimized laser cooling. In our current system, atoms are cooled to approximately 7 μK in a 200 μK trap through a combination of polarization gradient cooling (PGC) and adiabatic lowering of the trap depth, resulting in axial and radial position spreads of approximately 310 nm and 68 nm, respectively. Building on our previous work [54], further implementation of Raman sideband cooling (RSC) could cool the atoms close to their motional ground states, reducing their spatial distributions to approximately 66 nm (axial) and 31 nm (radial). Under these conditions, the aperture diameter can be increased by a factor of 2.5 without compromising coherence or gate fidelity, thereby reducing the Raman beam power loss from 80% to approximately 25%. In addition, since the Raman beams for different fiber channels can be supplied by separate lasers in our architecture, the laser source is not a limiting factor for scalability.

Revision:

In response to Reviewer 1's comment and in consideration of Reviewer 2's suggestion, we have moved the technical details regarding the use of an aperture-integrated filter for reducing the NA of the addressing beam to the *Polarization Control and Addressing Beam Uniformity Optimization* section of the Supplementary Information. We have also added a quantitative evaluation and discussion of the associated Raman power loss introduced by this approach, which is presented in the same section.

In the Supplementary Information, we **added** a new section titled *Polarization control and addressing beam uniformity optimization*.

In the *Polarization Control and Addressing Beam Uniformity Optimization* section of the Supplementary Information (page 2), we **added** evaluation and discussion of the Raman power loss, begins as follows (see Supplementary Information for full text):

“While this aperture-integrated filter improves the spatial uniformity of the addressing light experienced by the atoms, it also introduces power loss for the Raman beam. Experimental measurements in our current setup indicate that the 1 mm aperture leads to approximately 80% power loss of the Raman beam. Nevertheless, Raman power utilization can be significantly improved by relaxing the aperture constraint via optimized laser cooling.....”

Q2. The authors' current system does not seem to feature a way to resort an 55% filled array into a dense array of atoms. While this is not crucial for the current demonstration, it would be necessary to scale up.

Response:

We thank the reviewer for raising the issue of atom rearrangement, which has positively contributed to system improvement.

To address this concern, we have conducted additional atom rearrangement experiments. We have incorporated an atom rearrangement module into our system, which consists

of a spatial light modulator (SLM) and an acousto-optic deflector (AOD). The SLM generates a 5×4 optical tweezer array to capture reservoir atoms, while the AOD produces a steerable optical tweezer for transferring them into initially unoccupied fiber traps. The experimental details are illustrated in the revised Figure 1. With this configuration, we successfully prepared defect-free arrays of single atoms in 10 fiber traps, with a single-shot success probability of approximately 76%, far exceeding the 0.2% probability from stochastic loading. These results demonstrate the system’s scalability for high-density atomic array preparation. Moreover, the rearrangement efficiency can be further improved through enhancements in initial loading probability, transfer efficiency, and trap lifetime.

Revision:

We have added a new description in the *Scheme* section of the main text and in the *Atom rearrangement* subsection of the revised *Methods* section, detailing the implementation of defect-free atom array preparation in fiber array traps. Accordingly, we have updated *Figure 1* to include the atom rearrangement module in the experimental schematic, and replaced the previously shown averaged fluorescence image with a single-shot image demonstrating successful rearrangement.

In the caption of Fig. 1a (page 2), we **added**:

“The system includes a steerable optical tweezer generated by an acousto-optic deflector (AOD) and two static optical tweezer arrays: one created by an optical fiber array, and the other by a spatial light modulator (SLM). The SLM-generated traps serve as a reservoir for atom rearrangement, whereas the fiber-array-generated traps act as the main register for qubit operations.”

In the caption of Fig. 1c (page 2), we **changed**:

“Averaged fluorescence images of the single-atom array, taken with an exposure time of 50 ms.” **to** “Single-shot fluorescence image of an atom array with 50 ms exposure time. Left: Randomly loaded array. Red circles indicate the positions of the fiber traps. Right: Defect-free array obtained after atom rearrangement, with the SLM traps turned off prior to the second exposure.”

In the last paragraph of the *Scheme* subsection (page 3), we **changed**:

“Ten individual ⁸⁷Rb atoms are successfully loaded from a 3D-MOT into these optical tweezers, with a loading rate of 0.55, Fig. 1c and Fig. 1d.” **to** “Individual ⁸⁷Rb atoms are randomly loaded from a 3D-MOT into two static optical tweezer arrays, one generated by an SLM and the other by a fiber array, with an average loading probability of 0.55. These atoms are subsequently rearranged using a steerable tweezer into a defect-free array aligned with the 10 fiber traps, achieving a single-shot success probability of 0.76, FIG. 1c.”

In the revised *Methods* section (page 6), we **added** a new subsection titled *Atom rearrangement*.

Q3. A problem of this approach, which the authors recognise lies in the scalability arguably cold-atom based quantum computation will need to approach 10,000-100,000 atoms to achieve fault tolerant quantum computation. It does not seem feasible to build optical modules for each channel with bulk components at that scale.

Response:

We thank the reviewer for highlighting this important concern regarding scalability, which is indeed one of the central issues that we plan to tackle in our future work.

Based on our thorough evaluation, the optical modules used in this work are compatible with integrated photonics, making it feasible to scale the system to tens of thousands of channels. In our approach, the optical modules are used for beam splitting, intensity and phase modulation, and combining laser beams of different wavelengths. Beam combining can be scaled efficiently using fiber-based wavelength-division multiplexing (WDM), while other functions can be implemented using well-established on-chip photonic integration solutions [57,58]. For instance, the thin-film lithium niobate (TFLN) platform has been widely employed for multi-channel beam splitting, as well as electro-optic amplitude and phase modulation, all integrated on a single chip [59,60]. TFLN-based chips now offer performance comparable to bulk optics, while being significantly smaller in size and more energy efficient [61,62]. Customized fabrication of such chips is already commercially available from several companies. In our future work, we plan to adopt such technologies to enhance the scalability of our system.

Revision:

In response to the reviewer's comments, we have rewritten the relevant content in the *Discussion* section of the main text regarding the scalability of the system, and added supporting references to strengthen our arguments.

We **rewrote** the last paragraph of the *Discussion* section (page 6), the revised paragraph begins as follows (see revised manuscript for full text):

“Admittedly, this proof-of-principle demonstration is carried out on a small scale, the number of qubits can be increased by simply replicating the optical modules at the input end of the fiber array. In our current setup, these modules are built using bulk components. However, to reach the scale required for fault-tolerant quantum computing, it is essential to adopt well-established and scalable integrated photonics solutions [58, 59]. For instance.....”

Q4. A further problem may arise when we consider two-qubit gates: For high fidelity blockade gates [1] the separation between atoms needs to be on the order of a few um. It is not entirely clear that very high fidelity blockade gates together with single qubit gates can easily be realised in a static, equidistant array, since for single qubit gates the beam diameter in the atom plane has to be large compared with the trap size to reduce intensity variation induced error. For the narrow atom spacing required for high fidelity blockade gates this would lead to cross-talk.

[1]Evered, Simon J., et al. "High-fidelity parallel entangling gates on a neutral-atom quantum computer." *Nature* 622.7982 (2023): 268-272.

Response:

We thank the reviewer for the insightful comments regarding the compatibility between high-fidelity gate operations and low crosstalk in our system.

As demonstrated in a recent experiment [22], it is possible to implement high-fidelity two-qubit gates in a static, equidistant atomic array with interatomic spacing similar to that of our system, while maintaining low crosstalk. Specifically, with a spacing of 6 μm and an interaction strength of $2\pi \times 10.3$ MHz, a CZ gate fidelity of 0.9973(3) was realized, along with a single-qubit addressing crosstalk error as low as 1.5×10^{-5} [22], which is comparable to the crosstalk level observed in our current system. The dual-species architecture proposed and demonstrated in our previous work can further help mitigate crosstalk [36].

In addition, we have included preliminary experiments on Rydberg interactions between two atoms. Further optimization efforts—such as background electric field compensation and lowering atomic temperature—to enhance the fidelity of two-qubit gates are in progress, and the results will be reported in future work.

Revision:

We have added a new subsection titled *Rydberg blockade between two individual atoms* at the end of the Results section to describe the supplementary Rydberg excitation experiments. The results show a clear Rydberg blockade effect between neighboring atoms, indicating that our system provides a solid physical foundation for realizing high-fidelity two-qubit gates. In addition, we have quantitatively evaluated the crosstalk in single-qubit gate operations in the section *Optical crosstalk* of the Supplementary Information, and discussed its potential for further optimization.

In the *Results* section (page 5), we **added** a new figure, presented as Fig. 5.

In the *Results* section (page 5), we **added** a new subsection titled *Rydberg blockade between two individual atoms*.

In the section *Optical crosstalk* of the Supplementary Information (page 2), we **added** evaluation and discussion of the crosstalk, begins as follows (see Supplementary Information for full text):

“The maximum observed Rabi rate crosstalk η is approximately 1%, with an average value $\bar{\eta}$ of about 0.2%. Based on the single-qubit gate error model induced by crosstalk [5],

$$\epsilon_{\phi} = \frac{3}{2} \sin^2 \left(\frac{\eta\phi}{2} \right),$$

we estimate that for a π pulse ($\phi = \pi$), the corresponding gate errors are approximately 1.6×10^{-4} at maximum crosstalk and 6.6×10^{-6} on average.

In addition to perturbing the quantum evolution of spectator qubits, this crosstalk also

degrades the performance of parallel single-qubit gate operations. It originates from spatial interference between addressing beams.....”

For these concerns I am not sure that this work reaches the threshold for publication in Nature Communications.

Response:

We sincerely hope that our replies above have adequately addressed the reviewer’s concerns.

Reviewer #2 (Remarks to the Author):

In this article, the author introduced an innovative approach to implementing the single qubit gate using a fiber array architecture. The study addresses key challenges in achieving efficient and parallelized quantum operations on individual quantum bits (qubits).

In their experiment, the author has experimentally implemented successfully to load 10 Rb atoms into a fiber-generated optical tweezer, and achieve an average gate fidelity of 99.66%. A key advantage is that the parallel operation of the atom qubit in the setting. They have demonstrated, with an example, the simultaneous single-qubit operations on four randomly selected qubits, with an average gate fidelity of 99.61%. This result highlights the potential for scaling up quantum operations to larger systems. The success of this research opens new avenues for advancing quantum computing technologies, particularly in achieving high-fidelity gate operations on individual qubits and parallel gate operations. Therefore I think the work can be published in NC.

Response:

We sincerely thank the reviewer for their high recognition and positive evaluation of our work.

I have a few comments.

Q1. In the experiment, the author has focused on the single qubit operation. It is nice to see the high fidelity single qubit operation. I wonder why there is no demonstration of the entangled gates (such as two qubit control gates). Will the interference of the laser affect the performance of two qubit gates?

Response:

We thank the reviewer for raising this important point.

In the original manuscript, we mainly focused on demonstrating independent control of atoms enabled by the fiber array architecture. In the revised manuscript, we have added Rydberg excitation experiments, in which we successfully prepare an entangled state involving the ground and Rydberg states of two atoms—a key prerequisite for realizing high-fidelity two-qubit gates.

Interference between addressing beams does impact two-qubit gate fidelity. A recent experiment [22], when implementing high-fidelity two-qubit gates, encountered a similar issue. They addressed this by introducing a large frequency offset between the beams to average out interference-induced fluctuations. While this strategy could also be applied in our system, it would increase complexity. An alternative approach is to reduce the atomic temperature via Raman sideband cooling (as demonstrated in our previous work [54]), which allows the addressing beam size to be reduced by a factor of approximately 2.5, thereby suppressing interference-induced fluctuations to a level compatible with high-fidelity two-qubit gates.

Revision:

We have added a new subsection titled *Rydberg blockade between two individual atoms* at the end of the *Results* section to describe the supplementary Rydberg excitation experiments. The results show a clear Rydberg blockade effect between neighboring atoms, indicating that our system provides a robust basis for implementing high-fidelity two-qubit gates.

In the *Results* section (page 5), we **added** a new figure, **presented as Fig. 5**.

In the *Results* section (page 5), we **added** a new subsection titled *Rydberg blockade between two individual atoms*.

Q2. Towards the end of the “Scheme” section, the author claimed that the experiment achieved a loading rate 0.55. What is the unit here?

Response:

We thank the reviewer for pointing out this wording error. The value “0.55” in our original manuscript refers to the loading probability of single atoms into individual traps, and not a time-based loading rate. We apologize for the incorrect phrasing and have corrected this error in the revised manuscript.

Revision:

We have corrected this error at the end of the *Scheme* section in the main text, where “loading rate” has been revised to “loading probability.”

In the last paragraph of the *Scheme* section (page 3), we **changed**:

“Ten individual ^{87}Rb atoms are successfully loaded from a 3D-MOT into these optical tweezers, with a loading rate of 0.55, Fig. 1c and Fig. 1d.” **to** “Individual ^{87}Rb atoms are randomly loaded from a 3D-MOT into two static optical tweezer arrays, one generated by an SLM and the other by a fiber array, with an average loading probability of 0.55. These atoms are subsequently rearranged using a steerable tweezer into a defect-free array aligned with the 10 fiber traps, achieving a single-shot success probability of 0.76, FIG. 1c.”

Q3. Although the results are very interesting, I feel that the Results section contains too many technical details. Some of them might be useful to understand the experiment. However the main aim is still to achieve single and potential multiple qubit gate operation. I would suggest to move some of the subtle details to the SM, such that readers can understand the important advantage of the article, as well as what can be done with this setting.

Response:

We thank the reviewer for the thoughtful suggestion. In the revised manuscript, we have moved several technical details from the main text to the *Methods* section and the Supplementary Information, and substantially revised the *Results* section to clarify the presentation and highlight the key advantages of our approach. These adjustments do not affect the integrity of the main results and are intended to enhance readability.

Revision:

We have revised the *Results* section to improve clarity and focus. Specifically, several technical details have been moved from the main text to the *Methods* section and the Supplementary Information. In particular, both the detailed experimental procedures and the method for evaluating single-qubit gate fidelity via Randomized Benchmarking have been relocated to the *Methods* section, while the descriptions of polarization control and addressing beam uniformity optimization, as well as the crosstalk analysis, have been moved to the Supplementary Information.

In the revised *Methods* section (page 6 and page 7), we **added** three new subsections to provide relevant experimental and technical details:

- *Experiment details*
- *Atom rearrangement*
- *Randomized benchmarking of single-qubit gates*

The entire original *Methods* section has been **moved** to the Supplementary Information.

In the Supplementary Information, we **added** a new section titled *Polarization control and addressing beam uniformity optimization*.

REVIEWER COMMENTS

Reviewer #1 (Remarks to the Author):

The authors have mostly addressed my concerns and have expanded the data taken significantly. I am happy to recommend publication.

Response:

We sincerely thank the reviewer for the positive feedback on our revised manuscript and for recommending publication.

I would further recommend to also add the following citation on lithium niobate integrated photonics: <https://www.nature.com/articles/s41467-024-55423-3>.

Response:

We thank the reviewer for this insightful suggestion. We have now included the recommended citation—Christen, I. et al. An integrated photonic engine for programmable atomic control, *Nature Communications* **16**, 82 (2025)—in the revised manuscript (refer to page 8, line 626). We believe this reference provides important support to our arguments.

Reviewer #2 (Remarks to the Author):

The authors have made substantial improvements in the revised manuscript. All my concerns have been successfully addressed. The authors have clarified the text, and rewritten the manuscript to improve the readability. As far as I can tell, the authors have also addressed the comments of the first referee. Additional discussion and materials have been added to the revised manuscript, which do not only clarify the experimental results, but also highlight the new achievements.

Among the changes, I am particularly surprised by the new section on the demonstration of the Rydberg blockade (in section Rydberg blockade between two individual atoms). It has been shown that Rydberg blockade can be realized with the fiber array architecture with a high fidelity. This perfectly reflects the key theme of the article, i.e. a new possible way to carry out quantum computing with the Rydberg atoms. This is a nice piece of work providing a new way to build neutral atom quantum computers. I believe this work has important and direct contributions to the quantum computing community. It also impacts widely to the study of quantum simulation, light-matter interaction, and quantum information sciences.

I strongly recommend the publication of this work in Nature Communications.

Response:

We sincerely thank the reviewer for the very positive and encouraging comments on our work and greatly appreciate the strong recommendation for publication.